# Clocks without “Time” in Entangled-State Experiments

**DOI:** 10.3390/e22040434

**Published:** 2020-04-11

**Authors:** F. Hadi Madjid, John M. Myers

**Affiliations:** 1Consultant, 82 Powers Road, Concord, MA 01742, USA; gailmadjid@comcast.net; 2Harvard School of Engineering and Applied Sciences, Cambridge, MA 02138, USA

**Keywords:** entangled light states, synchronization, agent, unpredictability

## Abstract

Entangled states of light exhibit measurable correlations between light detections at separated locations. These correlations are exploited in entangled-state quantum key distribution. To do so involves setting up and maintaining a rhythm of communication among clocks at separated locations. Here, we try to disentangle our thinking about clocks as used in actual experiments from theories of time, such as special relativity or general relativity, which already differ between each other. Special relativity intertwines the concept of time with a particular definition of the synchronization of clocks, which precludes synchronizing every clock to every other clock. General relativity imposes additional barriers to synchronization, barriers that invite seeking an alternative depending on any global concept of time. To this end, we focus on how clocks are actually used in some experimental situations. We show how working with clocks without worrying about time makes it possible to generalize some designs for quantum key distribution and also clarifies the need for alternatives to the special-relativistic definition of synchronization.

## 1. Introduction

When I am meeting a friend for lunch at noon, I think, conventionally, of my watch as “telling time”; however, for some purposes we prefer an alternative theory of clocks based not on any theory of time but on relations that link a reading of one clock at the transmission of a signal to a reading of another clock at the reception of that signal. This frees us to think of each clock as coming with a faster-slower lever that its user can manipulate, according to the user’s purpose. Such a theory is reported in [1]. Here we extend this theory for application to situations in which agents, typically called Alice and Bob, make use of entangled photon pairs.

As an arena in which to consider the use of clocks, experiments with pairs of entangled photons present are especially interesting, because they involve locations at which two or more agents make use of clocks while they operate detectors and have occasion to communicate with one another across propagation delays. The situation of multiple detections raises vexed questions of the interpretation of quantum mechanics. The literature is too extensive to review here, but we point to questions announced long ago but still worth attention, arising in relating quantum mechanics to spacetime [2,3]. Although these questions are unlikely to be resolved any time soon to the general satisfaction of quantum physicists [4], we have a particular contribution to offer. In 2005 we proved, within the mathematics of quantum theory, that no evidence expressed as probabilities of outcomes can ever determine a unique explanation in terms of quantum states and linear operators, so that any choice of an explanation within the framework of quantum physics involves reaching beyond logic [5]. Hence, logically, the choice of an explanation is unpredictable, thereby showing a drastic unpredictability in physics, above and beyond quantum uncertainty. Multiple theories are always logically possible, including theories about clocks.

We consider situations in which *symbol-handling agents*, linked by quantum as well as classical communications, make use of clocks, for example, the Alice and Bob that appear in descriptions of quantum cryptography. By symbols, we have in mind letters of an alphabet, or, more basically, bit strings. We think of an agent as acting in steps, one step after another. An agent has a memory and can communicate with other agents. Because the agent operates one step after another, the agent handles symbols sequentially. An agent’s sequence of symbols can include symbols written by the agent as well as classical bits received from other agents, and also, in the quantum context, symbols transmitted by other agents, reporting detections of photons.

A general purpose for the use of clocks is to establish relations between symbols transmitted and symbols received. More exactly, agents in communication with one another use clocks to relate symbols possessed by one agent with symbols possessed by another agent. For this purpose, the clocks have to be, in one sense or another, synchronized. Some other investigators have addressed synchronization in conjunction with the employment of entangled photon pairs. In the next two sections we review a few of their reports [6,7,8,9] to show how our point of view of focusing on clocks without worrying about time widens the potential applicability of some designs for quantum key distribution and clarifies the need for alternatives to the special-relativistic definition of synchronization.

If we work with a notion of a global time coordinate, we base our thinking on an assumption, whether that of classical physics or special relativity or general relativity. The assumption blocks some avenues of exploration that open if, in contrast to assuming a global time, we avail ourselves of freedoms to construct “local times” linked to whatever cyclic processes we choose or invent. We highlight some of these freedoms in the remarks that intersperse the next two sections.

## 2. Case Studies Involving Entangled Photon Pairs

We consider two uses of entangled photon pairs. One use is for quantum key distribution (QKD); the other use is to synchronize separated clocks. The two uses are related, because suitably managed clocks solve what we call the “sequence-ordering problem” that arises in quantum key distribution. In its most basic form, QKD aims to provide two agents, Alice and Bob, with a cryptographic key that is theoretically secure against undetected eavesdropping by Eve [10]. Alice and Bob make use of their key to communicate privately, that is, to communicate encrypted messages, unreadable by Eve.

For our first case, reported by Tittel et al. in 2000 [6], we discuss a QKD experiment in which a pulsed laser cyclically pumps a down-converting crystal repetitively with short light pulses, thereby generating entangled photon pairs, with some efficiency less than 1. Unlike cases that follow, this case pumps with a pulsed laser that imposes a rhythmic cycle of operation on the experiment. Whether that rhythm is tightly connected to a standard frequency as defined in the International System (SI) turns out to be irrelevant. Another striking feature of this experiment is the employment of photon pairs entangled in such a way that phase correlations matter [11,12,13]. A source *S* generates a sequence of weak light pulses, theoretically described as single photons, into an optical fiber. The optical fiber forks into two branches, and each pulse splits into weaker pulses, one propagating along each branch. After propagating along the branches, with one branch imposing an extra delay relative to the other branch, the pulses enter again into a single fiber, one pulse delayed relative to the other. In addition, one pulse is offset relative to the other by an adjustable phase increment ϕ. In theory, what emerges is a single photon consisting of a superposition of an earlier and a later photon. The light next passes through a down converter, out of which comes light explained as a quantum state consisting of a superposition of tensor products of a pair of “early photons” with a pair of “late photons”.
(1)$$|\psi \rangle =\frac{1}{\sqrt{2}}\left(|s\rangle |s\rangle +e^{i\phi }|l\rangle |l\rangle \right).$$

Here, |s〉 is the early state and |l〉 is the later state. The light next goes into a fork in the optical fiber, with one branch of the fork directing the light toward Alice while the other branch directs the light toward Bob. As the light reaches Alice it enters another branching and rejoining with unequal arms, providing an earlier and a later pulse, and with another controllable phase increment α. The light then goes into another fork in the light path with each branch of the fork leading to a sensitive light detector. Bob operates symmetrically.

The upshot is that Alice and Bob need to operate in a cycle inherited from the pump laser. This cycle has several distinct phases. (These phases of a cycle have nothing to do with the phase ϕ or α of the narrow-band light pulse.) Alice has not just one phase of a cycle for detection, but three distinct phases. She has phases for early detection, middle detection, and late detection: early if an early pulse from the source is registered as traveling the early branch of Alice; middle for a superposition of an early branch at the source followed by a late branch at Alice’s receiver or vice versa; and late if a late pulse from the source is registered as traveling the late branch of Alice’s receiver. Ideally, for each cycle, just one of her detectors registers a detection and, furthermore, registers that detection in just one of the three phases. Again, Bob operates the same way.

Although the experiment used a single laboratory clock to drive all three cycles, that is, the cycles of the source, of Alice, and of Bob, a more basic consideration is that the source *S* numbers its pulses and that Alice and Bob, by estimating transmission delays, number their cycles of reception to match what is transmitted from the source. If the source intersperses its photon pairs with strong light signals, the strong light signals that arrive at Alice can act as her local clock, and the same for Bob. This arrangement allows for Alice and Bob to be mobile, relative to the source and to each other. In the mobile extension of the experiment, because of varying propagation delays, the clocks of Alice, Bob, and the source will tick at rates that vary relative to, for example, a Global Positioning System time coordinate.

This experimental design, in which Alice’s and Bob’s clock ticks are defined by signals arriving form the source, illustrates a form of synchronization markedly distinct from that defined by Einstein in special relativity. We call this condition that meshes a phase of reception with the arrival of a pulse *logical synchronization*. As discussed in Reference [1], logical synchronization (but not Einstein synchronization) is possible for two clocks in relative motion. To maintain logical synchronization, Bob would take a running average of deviations of arriving pulses from the center of the desired phase and use this running average in a feedback loop to adjust the rate of his clock.

**Remark** **1.**
*1.* 
*Unlike the continuous wave (CW) operation soon to be discussed, this pulsed operation with its source-imposed cycle allows for gating of light detectors, thereby reducing spurious detections [14].*
*2.* 
*There is no need for the clock that steps the pulse laser to have any definite relation to a laboratory time standard.*



An alternative method of coordinating Alice’s and Bob’s detections of photon pairs is the use of time stamps, as will be discussed in the examples below.

## 3. Experiments with Continuous-Wave Pumping

Three more examples come from the laboratory of C. Kurtsiefer at the Centre for Quantum Technologies (CTQ) at the National University of Singapore. The first two of these three are for QKD, while the third concerns the use of photon pairs in synchronizing clocks used for other purposes. As in the example above, all three of the examples from the CTQ involve agents Alice and Bob, and all three have certain additional common features:
They all involve polarization-entangled photon pairs as short light pulses.The pairs are generated not in the rhythm established by a pulsed laser, but by use of a continuous-wave (CW) laser, so that the pair production is more or less a Poisson process with unpredictable durations between photon pairs.Alice and Bob have separate clocks that they must occasionally adjust.Alice records light detections in a sequence of records and includes a reading of her clock in each record of detection. Bob operates the same way.Records made by Alice and Bob are subjected to post-processing in order to coordinate comparisons of detections stemming from a common photon pair.The unpredictability of the durations between photon pairs plays an indispensable role.

### 3.1. A QKD Experiment in 2006

The first of the three examples comes from the report of Marcikic et al. [7]. In this experiment, Alice employs a CW laser to generate unpredictably spaced photon pairs. She records detections and attaches a reading of her clock to each detection record. Similarly, Bob records detections and attaches a reading of his clock to each of his detection records. In order to identify which records are to be compared with which, Bob transmits (classically) his sequence of clock readings to Alice, who then computes the correlation of her sequence with Bob’s sequence. The correlation is a function of an offset variable τ (shown explicitly in Equations (3) and (4) of [8]). Under the assumption that the offset in clock readings does not change, (as would be the case if the propagation delay is constant), the correlation integral has a sharp peak at a value of τ that corresponds to the reading of Bob’s clock at the detection of a photon less the reading of Alice’s clock at her detection of a photon from the same entangled pair. Given this offset, the appropriate comparisons of paired detections can be made. For this to work, of course, Alice’s and Bob’s clocks have to be both stable enough and close in frequency.

The slowly varying clock offset is used in a feedback loop to limit the drift of Bob’s clock relative to receptions from Alice; that is, the running offset is fed back to adjust the frequency of Bob’s clock relative to Alice’s transmissions. The same feedback loop actually allows for a more general operation, in which Bob and Alice can be in gentle relative motion.

**Remark** **2.**
*1.* 
*This method of determining the correspondence between Alice’s and Bob’s clock readings depends critically on the unpredictability of durations between paired-photon emissions. If, as in the preceding experiment of Tittel et al., the pump laser was periodically pulsed, the correlation would show periodic peaks and thus be useless for guiding the adjustment of Bob’s clock.*
*2.* 
*The irregular photon pairs can be taken as ticks that mark an unpredictable local time.*



### 3.2. A QKD Experiment in 2009

A second experiment from the CTQ was reported by C. Ho et al., “Clock synchronization by remote detection of correlated photon pairs” [8]. The authors report an experiment on quantum key distribution in which they show how to avoid some otherwise stringent requirements for hardware synchronization. In examining their use of clocks, we speak of Alice and her clock readings $t_{A,i}$ where Ho et al. speak of reference clock *A* and its readings $t_{i}$; we also speak of Bob and his clock readings $t_{B,j}$ where Ho et al. speak of reference clock B and its readings $t_{j}^{\prime }$. The experiment involves a continuous source *S* of entangled photon pairs, with one photon directed to Alice and the other to Bob. The advance over the design in the preceding experiment allows the use of less stable clocks, achieved by an iterative scheme of feedback to adjust not only the offset of Bob’s clock readings relative to Alice’s clock readings, but also to steer the relative frequency of the two clocks. Because of this feedback, the use of post-processing might be called “prompt post-processing”; that is, one cannot wait too long before using the post-processed correlation to adjust the clock’s relative frequencies.

**Remark** **3.**
*1.* 
*As in the preceding example, the method of using correlations to adjust the relative rate of Alice’s and Bob’s clocks allows for Alice and Bob to be in gentle relative motion.*
*2.* 
*Relative motion of the clocks precludes their satisfying the conditions of synchronization specified in special relativity [1] yet does not limit the precision with which the clock readings can be made to correspond.*



### 3.3. An Experiment on Clock Synchronization in 2018

The third experiment from the laboratory of Kursiefer that we discuss is concerned not with QKD but with clock synchronization per se. In “Symmetrical clock synchronization with time-correlated photon pairs” Lee et al. use the correlation technique to “demonstrate a point-to-point clock synchronization protocol based on bidirectionally exchanging photons produced in spontaneous parametric down conversion (SPDC)” [9]. By employing entangled photons, the authors offer security against some (not all) malicious attacks on synchronization procedures, based on the ability to test violations of Bell inequalities.

The experimental set up doubles that of Marcikic et al. As in that experiment, Alice employs a CW laser to generate unpredictably spaced photon pairs. She records detections and attaches a reading of her clock to each detection record. Similarly, Bob records detections and attaches a reading of his clock to each of his detection records. For this experiment, however, Bob also employs a CW laser, so the activity of generation of photon pairs and their transmission goes on in both directions. Suppose Alice generates a given photon pair and detects one photon at reading $t_{A}^{X}$ of her clock, and suppose that Bob detects the other photon of the pair at reading $t_{B}^{R}$ of his clock. (The superscript *X* is for ‘transmit,’ and *R* is for ‘receive’.) Then $t_{B}^{R}-t_{A}^{X}$ is, in the notation of Lee et al., $\tau_{AB}$. Going the other way around, from Bob to Alice, we have $t_{A}^{R}-t_{B}^{X}$ is $-\tau_{BA}$ (note the minus sign) in the notation of Lee et al. Again using “prompt post-processing” to compute timing correlations, and acknowledging that they “made the strong assumption that the photon propagation times in both directions were equal,” the authors show how two correlation peaks combine to generate the needed steering to bring about their form of synchronization.

**Remark** **4.**
*1.* 
*In special relativity synchronization of proper clock A to proper clock B, with both clocks fixed relative to some inertial frame, invokes an (idealized) light signal from A at $t_{A}$ which echoes off B at $t_{B}$ and returns to A at $t_{A}^{\prime }$, so as to satisfy, independent of $t_{B}$, the equations*
(2)$$t_{B}=\left(t_{A}^{\prime }+t_{A}\right)/2.$$

*An informative discussion of other definitions of synchronization, applicable to improper clocks and to clocks in a curved spacetime, is given by Perlick in [15].*
*2.* 
*The authors invoke what amounts to (Equation 2) to express what they mean by “synchronization.”*
*3.* 
*The authors demonstrate experimentally that their criterion is met to within experimental tolerances for several different lengths of fibers, from which they conclude that the design works for clocks in relative motion. However, if by synchronization the authors mean satisfying (Equation 2), this conclusion is not strictly correct because, as pointed out by Perlick, the synchronization relation (Equation 2) is not symmetric and cannot hold bidirectionally for clocks in relative motion.*
*4.* 
*We note that although the bidirectional synchronization that accords with (Equation 2) is ruled out for clocks in relative motion, another form of synchronization is possible, and it is precisely the logical synchronization required for the bi-directional communications of digital symbols, discussed in [1].*



## 4. Discussion

The habit of thinking that whatever happens must happen in space and time (or, relativistically, in spacetime) is widespread; nonetheless, we assert that there is an alternative, for we see spacetime as a mathematical construct, visible only on the blackboard, for instance in expressions involving Lorentz transforms. In spite of legions of experimental evidence that accord with these blackboard expressions, we distinguish spacetime as a concept expressed in formulas from a theory of clocks based on relating symbols transmitted to symbols received.

Our proof of the multiplicity of explanations of given evidence, mentioned in the introduction, led to the later demonstration of unpredictability in the behavior of physical devices used in experiments [1]. We predict that examples of such unpredictability will be visible in records obtainable from experiments on entangled light states, such as those above, in the form of steering commands sent to adjust the clocks of Alice and/or Bob.

Interest in entangled states was advanced by a series of experiments on violations of Bell inequalities, leading to a particularly clear experiment on entanglement of polarization in 1982 [16]. An experiment on entangled states requires a system with several agents working at separated locations. Both in setting up an experiment and during its operation, in principle each agent has occasion to update the quantum state by which that agent explains the operation of the whole system, that is, the system that includes other agents. We like to picture any agent’s quantum state as written into a file local to that agent, on the basis of evidence available at the moment to the agent. From this point of view, each agent’s quantum state for the system is “local” in the sense of belonging to that agent. It is to be noted that the quantum state as used by an agent to guide that agent’s local actions cannot sensibly be taken to be a description of reality as seen by an observer who does not participate in the business of the system. This view of a quantum state as local to an agent resolves what otherwise is the serious obstacle to relativistic quantum theory that was reported by Aharonov and Albert [2,3].

An agent’s clock provides the agent with clock readings; that is, Alice’s clock provides her with a “local time.” Similarly for Bob, but neither Alice nor Bob are provided by their clocks with any “global time coordinate.” Thus, in addition to the locality of quantum states, we have “local times, ” and, as illustrated above, there does not seem to be any absolute need to invoke any concept of “global spacetime.” This point is further elaborated in [1].

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
