# Peer review of "Clocks without “Time” in Entangled-State Experiments"

_entropy, 2020, doi:10.3390/e22040434_

Round 1

Reviewer 1 Report

The manuscript by F. Hadi Madjid and John M. Myers with the title "Tracks of the unpredictable in entangled-state experiments" discusses several issues related to the synchronization of clocks and their impact on the implementation of quantum key distribution protocols with entangled two-photon states.

The article summarizes several ideas and claims to question the common notion of spacetime. Further, vague references are made to the measurement problem or some version of it (lines 22-24). It is unclear to me if the authors are aware of the complexity of that topic and any research that has been published on this topic, as they do not put their work into the context of existing literature, as they mainly cite their own papers.
Some formulations like lines 34-41, claiming that the discussion on the reality of the wave function is "largely unnoticed" suggest that the authors are ignorant of the literature. This topic has been covered widely in publications, nowadays even in textbooks and popular science books.

The authors introduce the concept of "logical synchronization" which refers to the idea to synchronize the receiver and choice of measurement basis with the sending of the photons. In a system with a pulsed source of entangled photons, an active choice of basis and gated detectors, this is really an issue.

The presentation of these ideas and especially their importance for the notion of space and time are not corroborated further by any detailed discussion or calculation. The manuscript stays so vague until it discusses the example of an experiment that distributed entangled photon pairs (Tittel et al. PRL 2000 4737-4740), where this "logical synchronization" would be an issue in case one of the receivers would be moving.

The authors point out that there is a problem, but present neither a quantitative notion of this problem nor any possible solutions. The community is well aware of this problem and mainly uses continuous experiments for a variety of reasons.

The so far only experiment which distributed pairs of entangled photons from a satellite (Yin et al., Science Vol. 356, 6343, pp. 1140-1144) also used a pulsed laser from the satellite in order to establish a common clock as a reference. However, in this case of a continuous source, a "logical synchronization" is not necessary in real-time, as the corresponding pairs of photons can be identified in post-processing.

In many experiments, the drifts of the clock can be corrected directly using the temporal correlation function. I can refer to Caleb Ho et al 2009 New J. Phys. 11 045011. This is the common method to handle clock drifts in entanglement-based QKD experiments.

Summarizing, this manuscript fails to present any novel ideas or solutions.

Reviewer 2 Report

In the reviewed manuscript, the authors discuss tracking of unpredictable in experiments on space-like separated systems, in particular, in quantum key distribution protocols.

The manuscript contains many philosophical judgements on:  (i) the mathematical formalism of quantum physics; (ii) unpredictability "in the maintaining logical synchronization"; (iii)  agents “engaged in a system explicable by entangled states", and only one mathematical formula specifying the entangled pure state of two qubits to which the authors refer as the “wave function” though for an N-qubit quantum state this term is not applicable.

In my opinion, this paper is more suitable for submission to a journal on conceptual issues in physics. 

Author Response

Thank you for your comments. We took the need to better connect concepts to experiments seriously, and have reorganized the entire presentation accord- ingly; we revised the title to better express our focus on the role of clocks in experiments. It is now “Clocks without ‘time’ in entangled-state experiments”. We quote your comments below and give our response.

Comments of Reviewer 2 In the reviewed manuscript, the authors discuss tracking of unpredictable in experiments on space-like separated systems, in particular, in quantum key distribution protocols.

The manuscript contains many philosophical judgements on: (i) the mathe- matical formalism of quantum physics; (ii) unpredictability ”in the maintaining logical synchronization”; (iii) agents engaged in a system explicable by entan- gled states”, and only one mathematical formula specifying the entangled pure state of two qubits to which the authors refer as the wave function though for an N-qubit quantum state this term is not applicable.

In my opinion, this paper is more suitable for submission to a journal on conceptual issues in physics.

Response to comments: We agree that we need to better connect our dis- cussion of conceptual issues to actual experimental situations. In the revised manuscript, we emphasize those connections, in part by discussing three more experiments. Also, in the revised manuscript we replace the term “wave func- tion” by “quantum state”.

Reviewer 3 Report

Understanding “correlation” and tracking the “unpredictability” are basic problems in quantum physics. The authors tried to address these problems in entangled-state exper- iments. I am not good at this professional field, and I can not give the relevant special comments. However, I think this topic is an interesting one. I suggest it should be sent to an expert for more professional comments.

Author Response

Thank you for your comment. We quote it and give our response as follows

Comment of Reviewer 3: Understanding correlation and tracking the unpre- dictability are basic problems in quantum physics. The authors tried to address these problems in entangled-state experiments. I am not good at this profes- sional field, and I can not give the relevant special comments. However, I think this topic is an interesting one. I suggest it should be sent to an expert for more professional comments.

Response to comment: Thank you for your expression of interest.

Round 2

Reviewer 1 Report

After the last review round, the authors modified the manuscript by removing some of the vacuous phrases which were criticized before and by adding two more experiments to be discussed.
The idea is now to "clarify the need for alternatives to the special-relativistic definition of synchronization" (page 2, line 45-46), which till the end is not clarified. In fact, everything the authors present to be at odds with special relativity seems very much compatible.

In describing the experiments, the authors often make questionable statements. Examples:

page 2, line 63-64: It is not so much the phase correlation but rather the fact that the photons were created at the same time.

page 2, line 69-70: There is no single photon at this point. It is still a laser pulse with many photons.

The description of the experiments is often a bit awkward. This "branching and rejoining" is an unbalanced Mach-Zehnder interferometer and a standard technique to detect time-bin entanglement.

Page 3, line 81: I do not see how the scenario described here is in any way in conflict with special relativity. In fact, sending light signals to one or two observers in other reference frame is a standard way to explain special relativistic effects in textbooks.

page 4, line 129-130: No. In fact, not every pulse contains a photon pair for two reasons: The transmission channel is in general lossy and creates a poissonian distribution of the mean number of pairs detected. More importantly, there is no deterministic single photon source (yet), therefore the source itself creates a pair with a rather low probability. The number of pairs created per pulse, on the other hand is related to the number of photons in the pump pulse, which is Poissonian as well.
The system is tuned in such a way that only very few pulses contain a pair at all, because pulses with double pairs deteriorate the fidelity of the observed quantum state. In this paper, there is a correlation function of pairs from a pulsed experiment: Ursin et al. Nature Physics volume 3, pages 481–486 (2007), figure 2b. It is really no problem to use the arrival time and correlation function of pulsed experiments to synchronize the sub systems to each other.

I do not see how this paper would present any technical or conceptual novelty. I do not recommend it for publication.

Reviewer 2 Report

The authors have inserted the needed changes and I could recommend the manuscript for publication in Entropy.